# Impact of Prescribed Exercise on the Physical and Cognitive Health of Adults with Down Syndrome: The MinDSets Study

**DOI:** 10.3390/ijerph20237121

**Published:** 2023-11-29

**Authors:** Viviane Merzbach, Michael Ferrandino, Marie Gernigon, Jorge Marques Pinto, Adrian Scruton, Dan Gordon

**Affiliations:** 1Cambridge Centre for Sport & Exercise Sciences, Anglia Ruskin University, Cambridge CB1 1PT, UK; viviane.merzbach@aru.ac.uk (V.M.); michael.ferrandino@aru.ac.uk (M.F.); marie.gernigon@universite-paris-saclay.fr (M.G.); jorge.pinto@aru.ac.uk (J.M.P.); adrian.scruton@aru.ac.uk (A.S.); 2CIAMS, Université Paris-Saclay, CEDEX, 91405 Orsay, France; 3CIAMS, Université d’Orléans, 45067 Orléans, France

**Keywords:** Trisomy-21, physical activity, fitness, walking, cognition, intellectual disability, executive, decision-making, vigilance, memory

## Abstract

The duplication of chromosome 21, as evidenced in Down Syndrome (DS), has been linked to contraindications to health, such as chronotropic and respiratory incompetence, neuromuscular conditions, and impaired cognitive functioning. The purpose of this study was to examine the effects of eight weeks of prescribed exercise and/or cognitive training on the physical and cognitive health of adults with DS. Eighty-three participants (age 27.1 ± 8.0 years) across five continents participated. Physical fitness was assessed using a modified version of the six-minute walk test (6MWT), while cognitive and executive functions were assessed using the Corsi block test, the Sustained-Attention-To-Response Task (SART), and the Stroop task (STROOP). All were completed pre- and post-intervention. Participants were assigned to eight weeks of either exercise (EXE), 3 × 30 min of walking/jogging per week, cognitive training (COG) 6 × ~20 min per week, a combined group (COM), and a control group (CON) engaging in no intervention. 6MWT distance increased by 11.4% for EXE and 9.9% for COM (*p* < 0.05). For SART, there were positive significant interactions between the number of correct and incorrect responses from pre- to post-intervention when participants were asked to refrain from a response (NO-GO-trials) across all experimental groups (*p* < 0.05). There were positive significant interactions in the number of correct, incorrect, and timeout incompatible responses for STROOP in EXE, COG, and COM (*p* < 0.05). Walking generated a cognitive load attributed to heightened levels of vigilance and decision-making, suggesting that exercise should be adopted within the DS community to promote physical and cognitive well-being.

## 1. Introduction

Down syndrome (DS), often referred to as Trisomy-21, results from the presence of either the whole or part of an additional duplication of chromosome 21 [1]. Recent works have demonstrated that the countenance of this extra chromosome can be attributed to alterations in protein expression, which have been linked to changes in physiological, biochemical, anatomical, cognitive, and metabolic profiles [2]. Such characteristics include chronotropic incompetence [3], neuromuscular conditions [4], reduced lung function [5], immunological suppression [6], as well as impaired decision-making, verbal reasoning, processing, attention, and problem-solving [4,7]. The ramifications of these characteristics are poor health outcomes as well as diminished social skills [8].

Coupled with these poor all-cause health outcomes across the DS community, there is also strong evidence showing that, as a population, they do not meet the recommended minimum for daily physical activity (PA). Recommendations from both the US Department for Health and Human Services and the National Health Service (NHS) in the UK suggest that all adults, including those with a disability, should engage in at least 150 min of moderate-intensity exercise or 75 min of vigorous-intensity exercise per week. A recent study [9] showed that a group of American adults with DS engaged in just 10.1 ± 13.5 min of moderate-intensity activity coupled with 1.7 ± 9.8 min of vigorous activity per day. Indeed, they were shown to engage in 412.7 ± 216.6 min of sedentary time per day. The lack of engagement in PA within the DS community has been investigated [10] regarding confounding variables from which several barriers were highlighted, of which cognitive and executive function, underlying medical and physiological factors, and lack of support were considered profound.

A meta-analysis [11] of thirteen randomised controlled trials involving 556 participants from the DS community highlighted that following prescribed exercise interventions, cardiorespiratory fitness, evidenced by changes in both maximal oxygen uptake (VO_2max_) and heart rate max (HR_max_), exhibited significant responses that favoured the intervention over control. Those findings are further supported by an earlier study [12], which highlighted that people with DS experienced large effects in favour of the exercise intervention in relation to body mass and waist circumference. However, a Cochrane review [13] emphasised a lack of well-controlled studies examining the impact of exercise interventions on physical and psychosocial health in adults with DS. Indeed, the authors stated that of the initial 1954 articles identified, only 63 referred to exercise and DS, of which only three met the guidelines for inclusion in the review.

The benefits of exercise on mental health and cognitive function are well recognised in the general population, with guidelines from the World Health Organisation (WHO), American College of Sports Medicine (ACSM) and NHS in the UK all citing evidence to support the adoption of exercise as a promotor of positive mental health and well-being. The benefits on cognitive function are also well documented and have been shown to confer positive responses in relation to memory and learning [14,15], as well as providing a counteraction to the associated decline in mental capacity with age [16]. These benefits are not confined to adults, with recent works highlighting the benefits of prescribed physical education on children’s cognitive and executive function [17]. The benefits are both acute [18] following a single bout of exercise, and chronic, with generational responses attributed to epigenetic modifications of the frontal brain lobe plasticity [19]. Irrespective of the mechanisms, the benefits of exercise on cognitive function have translated into improved verbal, perceptual, and numerical skills accompanied by enhanced memory control and cognitive flexibility [20]. However, the benefits of exercise on cognitive performance have been shown to have a ceiling effect [21], suggesting that individuals with lower performance on executive function tasks can anticipate greater benefits from a single bout of exercise. Crucially, these cognitive traits have all been documented to be impaired in those with DS compared to able-bodied controls [22], reflecting reduced capacities for decision-making, verbal reasoning, and articulation.

There is a shortage of data revealing the potential benefits of exercise on cognitive and executive function within the DS population. Those studies addressing the impact of prescribed exercise have focused on psychosocial outcomes attributed to exercise [23] rather than reflecting direct effects on cognitive function. Furthermore, as Tsou (2020) and Andriolo (2005) state, these studies are confounded by a lack of clinical measures, sample sizes and adequate controls [2,13]. As such, the DS population presents a unique combination of comorbidities, reflecting poor underlying health coupled with a high prevalence of inactivity. Additionally, exercise provides a stimulus for both physical and cognitive development. Therefore, the MinDSets study was conducted to examine if applying a prescribed exercise regime in the form of an eight-week intervention could positively affect the cognitive, executive, and physical health of adults with DS. It is hypothesised that walking will promote meaningful gains in both physical and cognitive functioning following an eight-week walking intervention and that these gains will be greater than those for cognitive-based training alone.

## 2. Materials and Methods

This study was approved by the Anglia Ruskin University Research Ethics Committee (SES_Staff_19–25), with all data collection conducted in accordance with the guidelines established via the Declaration of Helsinki [24].

### 2.1. Study Design

Using a repeated-measures, matched-groups design, participants were assigned to one of four groups based on pre-intervention measures of cardiorespiratory fitness. The groups were categorised as follows. Participants in the exercise intervention group (EXE) completed eight weeks of cardiorespiratory exercise, defined as either walking or jogging three times a week for a period of 30 min per session. A second group was defined as cognitive training only (COG) and completed eight levels (~20 min) of cognitive and executive function exercises per session six times per week. A combined group (COM) completed both the cardiorespiratory and cognitive interventions, as highlighted before. A final group acted as the controls (CON) and were not exposed to any intervention across the eight-week study period. Participants who were in CON were given the option, upon completion of the pseudo-eight-week intervention period, to be assigned to an intervention group and continue the study.

### 2.2. Participants

A total of 120 participants were initially recruited for this study. To be accepted into the study, participants had to be 18 years or older, be classed as ambulant, be cleared by their medical practitioner to participate, and have access to a mobile phone and computer. Furthermore, with a high prevalence of visual and auditory impairments in the DS community, participants were excluded if they were unable to visualise information on computer and mobile/tablet screens or to listen to instructions/auditory cues. Additionally, all participants had to have access to a helper or caregiver who could provide support throughout the duration of the study. Of the initial number, 35 participants were withdrawn due to being under the age of 18, medical grounds, illness, lack of time, or simply because they stopped responding. A further two participants from COG were excluded from the final data analysis due to low adherence rates or not completing the entire programme. Participants were recruited using a media campaign across Canada, North America, and Europe, as well as via direct mailing from the Canadian Down Syndrome Society. Eighty-three participants were recruited from North America (*n* = 67), Europe (*n* = 8), Africa (*n* = 5), Asia (*n* = 2) and Australia (*n* = 1) and included in the final analysis, with a mean age of 27.1 ± 8.0 years of which there were 40 females and 43 males (see Table 1 for further participant details, including the information split by groups). Recruitment was conducted over a period of 12 months to ensure that the criteria for minimum sample sizes were met. Group sizes were determined using previously established criteria for walking interventions in people with DS [25]. Minimum group sample sizes were estimated to be 20, assuming a power (ß) of 80% and an alpha level of 0.05 [26].

### 2.3. Measurements

All participants were provided with a mobile monitoring tool (Fitbit Inspire 2), which was set to record key movement-based parameters, including steps completed, distances covered, speeds, and heart rate. These were dispatched with instructional videos on how to wear, use and maintain the device. Additionally, a research assistant was available via online support to help set up the monitors and connect them with the participants’ mobile phone devices. Participants in the COG and CON were instructed to record the pre- and post-intervention fitness test data, whilst participants in the other two groups (EXE and COM) were also asked to utilise these devices to collect all the PA completed in the allotted training sessions.

Cardiorespiratory fitness was estimated using the six-minute walk test (6MWT) protocol, adapted from Chen (2018) [27]. The premise of this test, which is a routine measure in clinical populations [28], is that the distance covered is a proxy-indicator of cardiorespiratory fitness, e.g., the greater the distance completed in six minutes, the better the classification of cardiorespiratory health. All participants were sent detailed instructions, including an example video of a 6MWT. Participants were encouraged to select an environment in their local area that was flat, free of obstacles and for which they were instructed to cover as much distance as possible in the allotted six-minute period. For this measurement, the helper/caregiver was asked to walk behind the participant, providing encouragement throughout, but was at no point to set the pace of effort by getting ahead or by the participant’s side. This protocol was completed twice before the intervention period to establish reliability and act as a familiarisation trial and immediately after the intervention. Participants were then matched and randomly assigned to a group based on the furthest distance walked in either of the pre-intervention attempts of the 6MWT and, therefore, the highest estimated cardiorespiratory fitness levels. The four participants who walked the furthest were allocated to one of the four different groups, then the next four fittest participants were again assigned to the four different groups, and so on. 

Participants in the EXE and COM groups also completed a modified Talk Test (mTT) based on the protocol of Foster (2018) as part of the pre- and post-intervention assessments [29]. Written instructions, including an example video of a mTT, were provided before the completion of the test. The mTT is designed to replicate a laboratory-based graded exercise test where pulmonary gas exchange variables are recorded to determine the gas exchange threshold (GET). GET reflects the transition from predominantly aerobic exercise to exercise that relies more on non-oxidative (anaerobic) metabolism. As with the 6MWT, participants and helpers were instructed to find a safe and flat environment to complete the mTT. The helpers were asked to pace the participant for the mTT. For the first three-minute stage, the helper was instructed to set a pace that would be a very slow, comfortable walking pace for the participant and to maintain the same speed as consistently as possible for the whole three-minute duration. The participant was then told to recite the ‘Happy Birthday’ song out loud twice in the final 30 s of the three-minute stage. This song was chosen as it is a globally referenced song that is easy to follow. The participant was then asked if they could recite the song comfortably; this was also judged by the caregiver. If the answer was “yes”, the helper increased their walking speed with the participant keeping up, and the process was repeated. The final workload (speed), at which the participant could comfortably recite the song, was defined as their GET.

Cognitive and executive functions were assessed pre- and post-intervention in all participants. Using the online platform PsyToolkit [30,31], participants completed the following tests: Corsi block test (CORSI), Sustained Attention to Response Task (SART), and Stroop Task (STROOP). Each cognitive test was preceded by written instructions followed by an instructional video with examples of how to complete the test. CORSI assesses short-term memory recall, for which the participant was presented with nine randomly placed coloured blocks on the screen. In a random order, two blocks would initially change colour from pink to yellow, and the participant’s task was to select the correct blocks in the correct order. If correct, the number of blocks increased to three and so on; if incorrect, a second try at the same difficulty level was allowed. If the second try was also incorrect, the previous correct trial was counted as the CORSI span number. The SART was used to assess information processing and vigilance. There were 225 experimental trials during which white digits from 1 to 9 (each number appeared 25 times with never the same number following itself) were shown for 250 ms in the centre of the black screen, followed by a mask (a white circle with a cross) presented for 900 ms. The participant was instructed to press the spacebar (GO-trial) as quickly as possible during the 900 ms where the mask was shown unless the digit was a 3, in which case, they should refrain from pressing the spacebar (NO-GO-trial). Participants were allowed 18 practice trials ahead of the experimental trials in the SART, which were omitted from the final data analysis. Automaticity, a reflection of processing speed and selection, was assessed using the STROOP. Participants were presented with a colour word (red, green, blue, or yellow) written in one of the four colours. They were instructed to press a key (‘r’ for red, ‘g’ for green, ‘b’ for blue and ‘y’ for yellow) on the keyboard as quickly as possible but according to the ink colour and not the meaning of the word. During compatible trials, the ink colour and meaning of the word are the same (e.g., the word “green” written in green ink); for incompatible trials, the ink colour and meaning of the word do not correspond (e.g., the word “green” written in red ink). Participants had to respond to the stimuli within 2000 ms; if no key was pressed, it was counted as a timeout. Participants had 20 practice trials, which were not included in the final data analysis, followed by 240 experimental trials. The number of compatible and incompatible trials varied, but about a quarter of the trials were compatible, with the remaining three quarters being incompatible trials. The test assessed the delay in reaction time between compatible and incompatible stimuli.

Activity during the exercise interventions was monitored using the same approach highlighted for the pre- and post-intervention measures. The intensity of these sessions was based on the workload from the mTT at which the GET was predicted. Participants and their helpers were instructed that the speed and effort of the training sessions should not exceed this level/speed to ensure that all participants would be exercising at a predominantly aerobic exercise intensity. The cognitive training exercises were provided via BrainHQ (Posit Science, San Francisco, CA, USA). Participants were presented with a series of eight games designed to promote cognitive and executive function. The difficulty of the games adjusted depending on how well the participant was doing. Initially, six games appeared as part of the training schedule. The participants could unlock the remaining two games once they had reached a certain difficulty level in the first six games. Games were selected following feedback provided by participants from a pilot study in a DS community (*n* = 12; five females and seven males; age 29.2 ± 4.9 years) based on accessibility and not being overly complex but stimulating. Additionally, the suite of games selected was adjusted if the participant presented with either a VI or AI. Games that relied more on auditory cues were excluded from the suite of games for participants with AI, whilst games that depended heavily on visual cues were excluded for participants with VI and replaced by games that were more suitable for the given impairment.

### 2.4. Data Acquisition

All participants were registered using a bespoke app accessible via a browser on their mobile phones (MinDSets, Bliss Innovative Maker Studio, Brussels, Belgium). The app was pilot-tested by individuals with DS (*n* = 12) to assess its accessibility and ease of use. It was designed using large icons, clear contrasts in colour and with minimal need for instructions. It served as a portal whereby the participants could be contacted individually and sent reminders of when activities needed to be completed. Additionally, the app acted as a data logger connecting to the personal Fitbit and BrainHQ accounts anytime these were activated. These datasets were then transferred to a spreadsheet for initial screening and to monitor for missing or erroneous data. MATLAB (MathWorks MATLAB R2023a, Natick, MA, USA) was used to align the datasets within and between groups for all measures. Data for CORSI is presented as the highest number of correct blocks selected (n), while for SART, data are shown as the mean response time (ms) and the number of correct and incorrect responses (n). STROOP data are extracted as percentages of correct, incorrect, and timeout responses for compatible and incompatible trials, along with the associated reaction times (ms).

### 2.5. Statistical Analysis

Using SPSS (IBM^®^ SPSS Statistics, version: 28.0.1.1 (15), Armonk, NY, USA), data are expressed, where appropriate, as mean ± SD. Where required for parametric evaluation, normality was first determined using Kolmogorov-Smirnov test. Shapiro-Wilk test was preferred for the mTT data (as *n* < 50). A follow-up test of homogeneity of variance (Levene’s test) was also conducted where appropriate. Differences between groups at baseline in physical and cognitive measures were assessed using one-way ANOVA with post-hoc testing evaluated using Tukey HSD; the Kruskal-Wallis test was applied for non-parametric data. Time-by-group interactions across all measures and groups for parametric data were determined using 2-way repeated-measures ANOVA. If Levene’s test was violated, Games-Howell post-hoc testing was preferred. For non-parametric data collected for this study, Wilcoxon signed rank tests were utilised to assess pre- to post-intervention measures within groups. Where baseline and post-intervention differences between groups were found, repeated-measures ANCOVA were applied to confirm if post-intervention differences were affected by baseline differences. Pearson Chi-square tests were used for the count and percentage data of SART and STROOP, respectively, to test for interactions from pre- to post-intervention within groups. An alpha level of *p* < 0.05 was considered statistically significant.

## 3. Results

### 3.1. Baseline

There were no significant differences in age, Raven’s advanced progressive matrices set 1 score (Table 1), self-reported PA levels (Table 2), CORSI span, SART measures, percentage of incorrect responses for compatible and incompatible trials in the STROOP, or the physical fitness measures (6 MWT and mTT) between the groups before the start of the intervention (*p* > 0.05). However, there were significant differences in the mean overall response times for correct compatible (F(3,78) = 4.467, *p* = 0.006, η^2^ = 0.147) and incompatible (F(3,75) = 3.029, *p* = 0.035, η^2^ = 0.108) responses in the STROOP. Post-hoc testing revealed that, on average, EXE responded 209.6 ms (*p* = 0.014) and COG 207.3 ms (*p* = 0.021) faster than CON participants for compatible trials. For incompatible trials, EXE were, on average, 198.8 ms faster than CON (*p* = 0.039). There were significant differences in the percentage of correct responses for compatible (H(3) = 9.645, *p* = 0.022) and incompatible (H(3) = 9.018, *p* = 0.029) trials, as well as significant differences in the percentage of timeouts for compatible (H(3) = 11.200, *p* = 0.011) and incompatible (H(3) = 9.826, *p* = 0.020) trials in STROOP.

### 3.2. Intervention Outcomes

Adherence rates for EXE, COG, and COM across the eight-week intervention period are shown in Table 3.

#### 3.2.1. Physical Fitness

Following the eight-week interventional period and independent of grouping, there was an overall significant increase in total distance covered in the 6MWT of 23.4 ± 88.0 m from 498.8 ± 101.3 to 522.1 ± 104.8 m (F(1) = 5.175, *p* = 0.026, η_p_^2^ = 0.065). There were also significant time-by-group improvements in 6MWT distance for EXE by 11.4%, equating to a gain of 55.6 ± 66.4 m (*p* = 0.003, η_p_^2^ = 0.113) and for COM showing a 9.9% increase in distance walked of 49.2 ± 64.7 m (*p* = 0.008, η_p_^2^ = 0.091) as shown in Figure 1. For COG and CON, 6MWT distance was not significantly different from pre- to post-intervention (*p* > 0.05). There were also non-significant changes in the distance covered during the fasted lap, where reciting ‘Happy Birthday’ was still possible, during the mTT for both EXE from 270.7 ± 47.5 to 272.6 ± 40.6 m and COM from 292.8 ± 65.0 to 308.6 ± 78.6 m (*p* > 0.05).

#### 3.2.2. Cognitive Responses

Figure 2 highlights that the CORSI span improved for all groups (EXE by 10.5%, COG by 15.6%, COM by 11.8%, and CON by 26.9%) but only significantly in CON (Z = −1.997, *p* = 0.046).

Overall, the response time for SART in the GO trials for correct responses improved, with participants responding 24.6 ± 86.5 ms faster after the eight-week intervention period compared to baseline measures (F(1) = 6.568, *p* = 0.012, η_p_^2^ = 0.077) as presented in Table 4. However, non-significant time-by-group associations were recorded in this measure (*p* > 0.05). Chi-square analysis highlighted positive significant interactions between the number of correct and incorrect responses from pre- to post-intervention for all groups in the GO-trials (total responses: Pearson Chi-Square (1) = 128.801, *p* < 0.001; EXE: Pearson Chi-Square (1) = 18.973, *p* < 0.001; COG: Pearson Chi-Square (1) = 59.563, *p* < 0.001; COM: Pearson Chi-Square (1) = 127.592, *p* < 0.001; CON: Pearson Chi-Square (1) = 6.865, *p* < 0.009) (Figure 3A and Table 4). For incorrect responses in the NO-GO trials, response times were recorded and showed significant speeding from pre- to post-intervention for the whole group by 22.3% (Z = −3.765, *p* < 0.001), for EXE by 24.2% (Z = −2.091, *p* = 0.037), and for COM by 21.1% (Z = −2.138, *p* = 0.033) shown in Table 4. There were also positive significant interactions between the number of correct and incorrect responses from baseline to post-intervention in the NO-GO-trials for total responses (Pearson Chi-Square (1) = 30.465, *p* < 0.001), EXE (Pearson Chi-Square (1) = 18.562, *p* < 0.001), COG (Pearson Chi-Square (1) = 7.907, *p* = 0.005), and COM (Pearson Chi-Square (1) = 4.580, *p* = 0.032). No significant interactions of the number of correct and incorrect responses were found for CON in the NO-GO-trials of the SART (Pearson Chi-Square (1) = 3.362, *p* = 0.067) (Figure 3B and Table 4).

For the compatible trials of the STROOP, there were significant improvements in correct response times for all participants from the initial test to the final assessment by 42.9 ± 187.0 ms (F(1) = 4.554, *p* = 0.036, η_p_^2^ = 0.055) as presented in Table 5. Furthermore, repeated-measures ANOVA highlighted significant differences between groups when assessing the combined average response times from pre- and post-intervention (F(3) = 5.881, *p* = 0.001, η_p_^2^ = 0.184). Post-hoc analysis demonstrated that EXE responded 188.6 ms faster (*p* = 0.011) and COG 227.4 ms faster (*p* = 0.002) during the compatible trials than CON. However, when baseline differences in correct response times for compatible trials between groups were considered, repeated-measures ANCOVA could not confirm any significant differences between groups (F(3) = 1.905, *p* = 0.136, η_p_^2^ = 0.069). Chi-square analysis showed positive significant interactions between the number of correct, incorrect, and timeout responses for compatible trials for total responses (Pearson Chi-Square (2) = 145.264, *p* < 0.001), EXE (Pearson Chi-Square (2) = 9.530, *p* = 0.009), COG (Pearson Chi-Square (2) = 92.360, *p* < 0.001), COM (Pearson Chi-Square (2) = 48.479, *p* < 0.001), and CON (Pearson Chi-Square (2) = 33.887, *p* < 0.001) (Figure 4A and Table 5). For incompatible trials in the STROOP, repeated-measures ANOVA revealed significant differences in response times between groups (F(3) = 3.549, *p* = 0.018, η_p_^2^ = 0.124), with post-hoc analysis displaying that participants in COG responded 198.0 ms faster than CON (*p* = 0.017). However, repeated-measures ANCOVA, with baseline differences between groups in response time taken into consideration, could not confirm the differences in response time for incompatible trials between COG and CON or any other group interaction (F(3) = 1.449, *p* = 0.236, η_p_^2^ = 0.055). Chi-square analysis highlighted positive significant interactions between the number of correct, incorrect, and timeout responses for incompatible trials for total responses (Pearson Chi-Square (2) = 310.424, *p* < 0.001), EXE (Pearson Chi-Square (2) = 25.794, *p* < 0.001), COG (Pearson Chi-Square (2) = 184.242, *p* < 0.001), COM (Pearson Chi-Square (2) = 180.508, *p* < 0.001), and CON (Pearson Chi-Square (2) = 47.292, *p* < 0.001) as shown in Figure 4B and Table 5.

## 4. Discussion

This study is the first of its kind using large sample sizes to examine if a period of prescribed exercise can positively affect the physical health as well as the cognitive and executive function of people with DS. We hypothesised that prescribed PA in the form of walking/jogging would act as a cerebral modulator by promoting key cognitive components of information processing, decision-making, and traits associated with executive function, such as pattern recognition. The findings allow for the acceptance of the hypothesis, as eight weeks of walking were shown to increase outcomes in physical health, information processing (SART) and selective attention (STROOP).

Adherence rates for both the exercise and cognitive interventions were excellent, with participants completing the requisite number of sessions with only a very minor number of sessions missed. Indeed, the adherence rates for the walking/jogging exercises were 108.1 ± 21.5% (EXE) and 110.3 ± 16.9% (COM), while for the cognitive training, they were between 95.5 ± 10.1% (COG) and 100.8 ± 12.4% (COM). These show a good level of fidelity in the data and are in accordance with previous works [32] that show adherence rates are high when there is a form of supervision to the intervention, as was applied in this study through the research team.

The 6MWT data at baseline across the four groups (498.8 ± 101.3 m) are comparable to data from adults with intellectual disabilities and DS (490.4 ± 58.9 m) [33], highlighting that the physical fitness status of the MinDSets study population was representative of the DS population. The changes observed in the 6MWT data for both EXE and COM groups highlight that as little as 24 sessions of low-intensity exercise were enough to confer a notable biological response, with the 6MWT representing a proxy-indicator for cardiorespiratory health.

Why should walking act as a cerebral modulator in this population? Walking is a complex task that needs a multifaceted activation of both cortical and sub-cortical brain areas [34], with both a direct and indirect locomotor (walking) pathway [35]. The direct pathway drives locomotion via activation of the cerebellum and the spinal cord, whilst the indirect locomotive pathway regulates the stability of movement via the basal ganglia, prefrontal cortex, and premotor areas [34,35].

Walking, amongst many activities that are performed daily, is referred to as a goal-directed action. These activities are almost sub-conscious for the able-bodied population and require very little attention to be paid to them [36] but necessitate a close dialogue between the perceptual and motor components of behaviour. Thus, it is theorised that the control and execution of the goal-directed task (in the case of this study, walking) relies upon what Montagne (2003) refers to as an information-movement cycle [36]. Accordingly, for every movement that is undertaken, an optic flow of information is generated; for every muscle action leading to the action of walking, information is generated in the form of optical invariants, which are continuously monitored to inform the individual about the validity of the execution of the task. Therefore, walking is a task that applies a cognitive load, albeit minimal, in the able-bodied population.

However, walking acts as a pivotal stimulator of cognitive processing in individuals with DS, where PA levels are lower than in the general population and where there are impaired cognitive processing capabilities [9]. Although not assessed in this study, previous works have shown profound issues of coordination and motor control within the DS community, which decline further with age [37]. In the context of the increased scores for selective attention, vigilance, and information processing as reflected via the outcomes for the SART and STROOP tests, it is hypothesised that walking acted as a cerebral modulator by stimulating the direct and indirect locomotor pathways.

Walking required those in the EXE and COM groups to pay attention to the task at hand, triggering the information-movement cycle. The simple act of walking necessitated that the participants became more vigilant in terms of what they were doing, as shown in the outcome of the SART test as a function of the optic flow. Indeed, interrogation of the SART data reinforces this argument further. The test is composed of GO- and NO-GO-trials, both reflecting levels of vigilance and information processing. In the GO-trials condition, irrespective of the number that appears on the screen aside from the number 3, the participant just needs to tap the spacebar. Whereas for a NO-GO condition, greater levels of vigilance are required as participants must now refrain from pressing the spacebar when the number 3 appears. Those in the EXE group exhibited the lowest contribution of incorrect attempts to the overall responses in the NO-GO condition post-intervention, highlighting that the walking promoted a reduced state of what Jackson termed a ‘wandering mind’, fostering heightened levels of vigilance [38]. Although walking is only considered to have a low cognitive load in the able-bodied population, it would appear to provide a heightened load in the DS population, fostering the engagement of these information processing cycles and thereby raising the cognitive load.

The STROOP data provides a unique insight into the decision-making and selective attention capabilities of an individual, as reflected via the outcomes for both the compatible and incompatible trials. Additionally, this test provides some further insight into the speed at which these decisions can be rendered. In a compatible condition, the word and ink colour presented would match, while in an incompatible condition, these do not match, and the ink colour must be indicated rather than the meaning of the word. Of note, at baseline, there were significant differences in timeout responses in the compatible trials between groups, with COM and CON having the highest contribution of timeout responses to overall responses when compared to both EXE and COG. This indicates an increased degree of uncertainty and indecisiveness, key modulators of decision-making and attention [39]. Participants in the COG and COM groups showed the greatest decrease in timeout responses when comparing pre- to post-intervention. Both groups were exposed to games that actively promoted making decisions. However, the notion of walking fostering heightened decision-making attributes is reinforced using the reflection of the incompatible conditions, where the EXE group had the highest percentage of correct responses both before and after the intervention compared to all other groups. Those in the CON group exhibited the highest contribution of both incorrect and timeout responses to their overall results, with minimal changes from pre- to post-intervention in these outcomes. From this, it can be postulated that the CON group, who were not exposed to an intervention of either walking or cognitive training, did not engage in decision-making processing during the eight-week period beyond their usual daily exposure.

Furthermore, recent works have shown, using functional magnetic resonance imaging, that decision-making is a dual process reflecting both the accuracy of the decision and the speed [40]. This is important in the context of this study, as walking has been shown to promote decision-making attributes (accuracy) but not the speed at which these are reached [41]. This is reflected in the EXE group, where the response times either did not change (compatible trials) or even slowed down (incompatible); both outcomes were non-significant. In contrast, those in the COG and COM groups were exposed to games that promoted the development in both the accuracy and speed of making a decision, as reflected in the higher percentages of correct responses as well as speeding in response times, albeit non-significant for both groups, in both compatible and incompatible trials.

Indeed, the proposed hypothesis is reinforced by the findings that those in the COM group exhibited a magnified response in both SART and STROOP outcomes when compared to the EXE and COG groups alone. The performance in the COG group was perhaps not surprising as the games selected in the intervention training focused on the critical attributes of cognitive and executive function, fostering development in learning to learn, decision-making and visuospatial awareness. The findings from this study are in broad agreement with previous works, which have shown that a period of ‘mental skills’ training can improve performance in cognitive task execution in a DS population [42]. Thus, in the COM group, where the participants were exposed to both cognitive training six times per week and walking three times per week, there were more significant gains in both SART and STROOP responses when compared to the COG and EXE groups alone.

A finding that is perhaps more surprising is that there were noteworthy, albeit non-significant, increases in performance in the CORSI test for EXE, COG and COM. Unlike the SART and STROOP tests, the CORSI test assesses working short-term memory connected to visuospatial awareness [43]. The connection to exercise regarding working memory has been established previously in a non-DS population [20]. In this meta-analysis, the authors concluded that a single bout of aerobic exercise conferred positive responses in executive function, with an emphasis on working memory. Studies of working memory have suggested that a key facet in its development is attention and, by extension, distraction [43].

Thus, it is likely that the previously cited heightened performance in the SART test for EXE, COG, and COM, via activation of the direct and indirect locomotor pathways, would also trigger modulations in short-term memory through increasing vigilance and attention-orientated goals. This is a finding that is reinforced by the changes from pre- to post-intervention for the CORSI test and the number of correct responses recorded in both the GO- and NO-GO-trials in the SART test. Though the significant positive change in the CORSI test for CON is intriguing and is something that needs further consideration in future works, there are some suggested explanations for this finding. Potentially, there could have been an enhanced learning effect in CON in the post-intervention measure. A third of the participants in this group scored a 0 for the Corsi span in the pre-intervention assessment, compared to only 13.6% in EXE, 10.5% in COG, and 18.2% in COM, allowing for a greater potential improvement when the CORSI was completed for the second time post-intervention. Furthermore, participants in CON reported the lowest weekly sedentary time (Table 2), whilst the sedentary time in the other three groups was higher and comparable to previously reported data [9]. Self-reported moderate and vigorous PA time was also the highest in CON, with 81.0% of the CON participants also engaging in sports, fitness, or recreational activities (EXE 72.7%, COG 78.9%, and COM 81.0%). Although it has been suggested that self-reported levels of PA tend to be overestimations [44], it can be speculated that participants in CON were more active before and potentially during the MinDSets study, which could have caused the results in the CORSI test.

Of note, there are several limitations associated with this study. The primary limitation relates to the mode of data collection for all variables. As this was a global study with participants across five continents, approaches had to be adopted to allow for remote data collection that would still render reliable and valid data. As such, the helpers and caregivers became ‘citizen scientists’ who had to administer the cognitive tests, calibrate the Fitbit devices, and ensure that walking speeds were replicable and that all training sessions were completed. Such an approach, although increasing the ecological validity of the study, does mean that more robust and clinically worthwhile measures were not possible. Coupled with this, there is recognition that the eight-week intervention is restricted in duration and only provides limited insight into the degree of modulation that can be provided through exercise and cognitive training. Furthermore, parts of the data collection were conducted during winter with poor weather conditions (very low temperatures and high levels of snow on the ground). These conditions could have impacted the changes in the pre- to post-intervention fitness assessments for affected participants with lower scores in the post-measures due to worse weather conditions, as well as the intervention training itself. Results of the CORSI should be interpreted with caution as there was no familiarisation trial programmed into the assessment. Participants only had written instructions and an example video of how to complete the CORSI. The authors also recognise that there is variability in some of the cognitive measures at baseline, a reflection of the study design and matching of groups on physical fitness from the 6MWT data. Upon reflection, the groups may have benefited from being matched using an amalgamation of estimated fitness levels and pre-intervention cognitive measures.

The findings from this study should now provide impetus for future work. Although walking yielded significant cognitive gains, it is compared to other modes of PA, relatively simple in its application. Presenting a more complex locomotive activity may foster greater improvements in physical and cognitive development. Coupled with this, previous works have indicated that the magnitude of cognitive development in a non-DS population is a function of exercise intensity [45]. In the DS population, little is known about how exercise intensity drives biological responses and, crucially, if the cognitive developments observed in this study could be heightened through exercise, which is more intense. It is also recommended that future works should apply intervention periods beyond eight weeks to establish the longitudinal responses.

## 5. Conclusions

This is the first study of its kind to examine if applying a period of prescribed exercise can positively affect physical and cognitive health in a DS population. The findings are significant and offer a crucial challenge to the DS and wider societies. Through the simple application of walking, a form of exercise which requires little to no equipment or expense, there were significant increases in cognitive and executive function, reflecting improved capabilities in key attributes of information processing, vigilance, and selective attention. These responses were magnified through a combined dose of exercise and cognitive training and offered a real scenario that can be adopted with the DS community for increasing cognitive functioning. The ramifications of these findings are significant for the DS community. Increased cognitive function will help foster increased societal integration and quality of life, which, given that this is the first generation of those with DS to outlive their parents and caregivers, is of importance.

## Figures and Tables

**Figure 1 ijerph-20-07121-f001:**
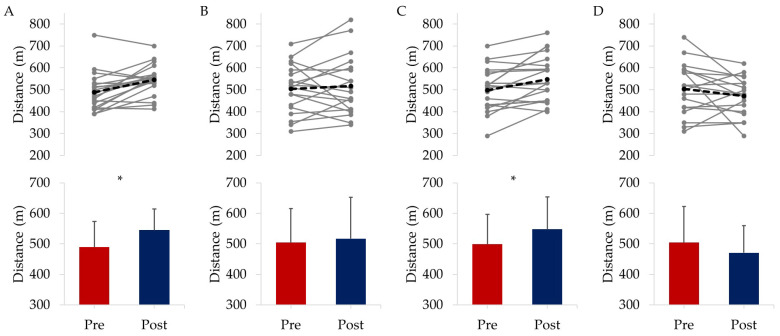
6MWT distance where (**A**) = EXE, (**B**) = COG, (**C**) = COM, (**D**) = CON. The top row panels show individual responses with the mean shown as a dashed black line, while the bottom row panels display the mean ± SD for each group with * highlighting significant differences from baseline to post-intervention.

**Figure 2 ijerph-20-07121-f002:**
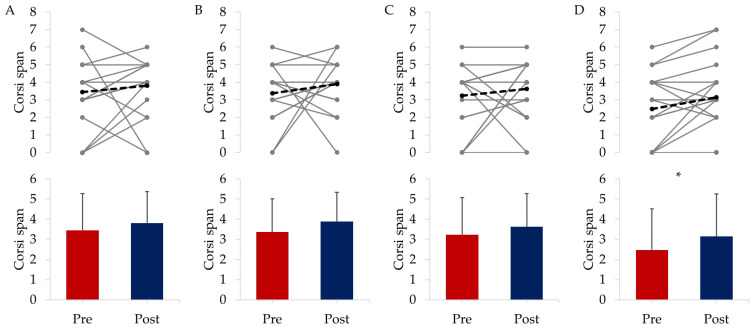
CORSI span where (**A**) = EXE, (**B**) = COG, (**C**) = COM, (**D**) = CON. The top row panels show individual responses with the mean shown as a dashed black line, while the bottom row panels display the mean ± SD for each group with * highlighting significant differences from baseline to post-intervention.

**Figure 3 ijerph-20-07121-f003:**
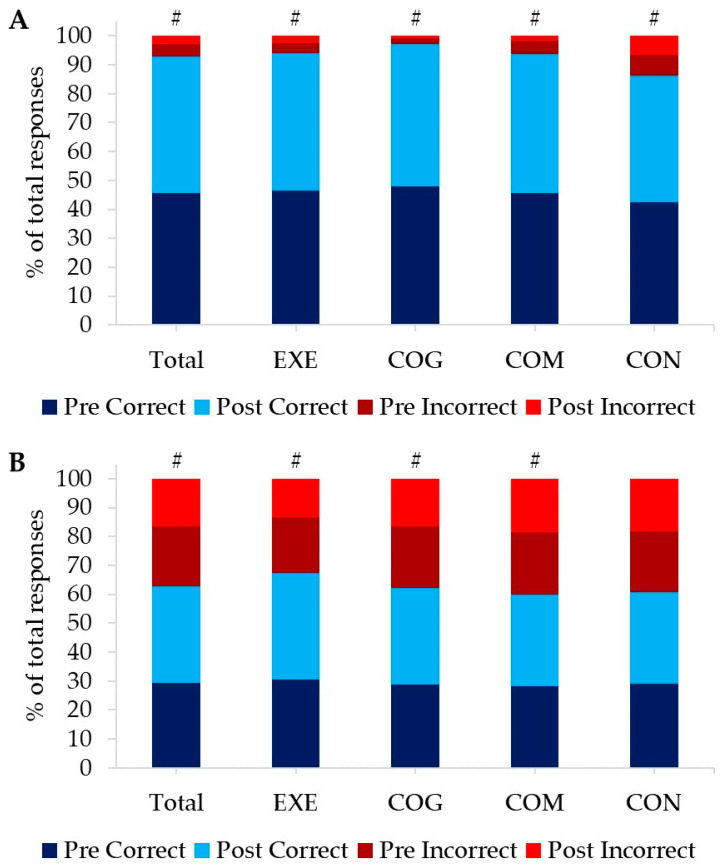
SART outcomes—Percentage contribution to total responses of correct and incorrect responses for pre- and post-intervention in the GO-Trials (**A**) and NO-GO-Trials (**B**) for all participants combined (Total) and split by groups. Positive significant interactions from Chi-square analysis are highlighted with #.

**Figure 4 ijerph-20-07121-f004:**
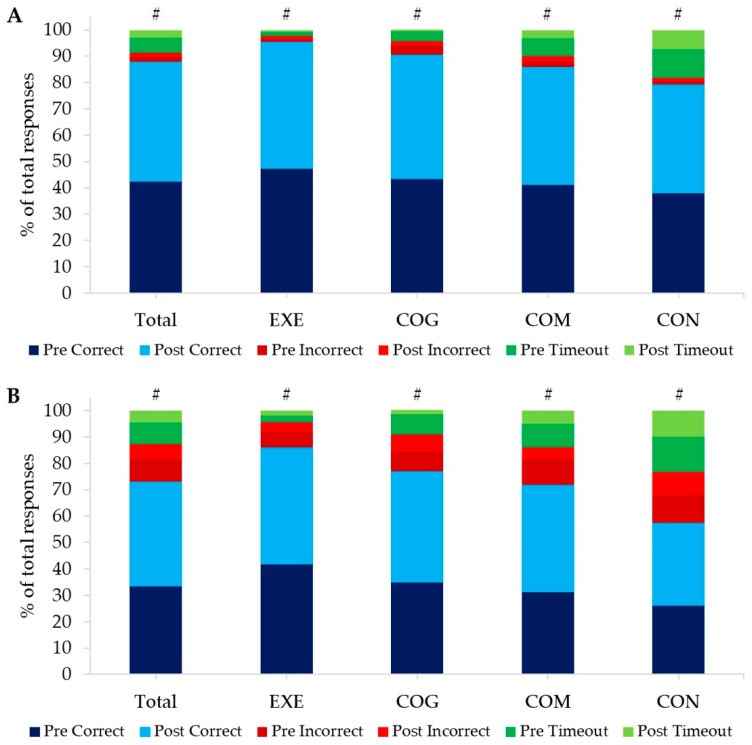
STROOP outcomes—Percentage contribution to total responses of correct, incorrect, and timeout responses for pre- and post-intervention in compatible (**A**) and incompatible (**B**) trials in the STROOP for all participants combined (Total) and split by groups. Positive significant interactions from Chi-square analysis are highlighted with #.

**Table 1 ijerph-20-07121-t001:** Participant characteristics for the whole cohort and split by group, including details about visual and auditory impairments and the Raven’s Advanced Progressive Matrices Scores.

	Total	EXE	COG	COM	CON
**Number of participants**	**83**	**22**	**19**	**21**	**21**
**Males**	43	9	4	14	16
**Females**	40	13	15	7	5
**Age (years)**	27.1 ± 8.0	25.0 ± 7.1	26.2 ± 6.1	28.9 ± 9.2	28.2 ± 9.0
**Countries**	Canada (53) Finland (1) Germany (1) Ireland (2) Myanmar (1) New Zealand (1) South Africa (5) Thailand (1) UK (4) USA (14)	Canada (14) Finland (1) Ireland (1) South Africa (1) Thailand (1) UK (1) USA (3)	Canada (12) Ireland (1) Myanmar (1) South Africa (2) UK (1) USA (2)	Canada (13) Germany (1) New Zealand (1) South Africa (1) UK (1) USA (4)	Canada (14) South Africa (1) UK (1) USA (5)
**Visual impairment (VI)** *****	**24** Amblyopia (2) Astigmatism (10) Cortical visual impairment (1) Essential iris atrophy (1) Hyperopia (8) Keratoconus (1) Left eye blindness (1) Myopia (8) Nystagmus (2) Strabismus (3)	**6** Amblyopia (1) Astigmatism (2) Cortical visual impairment (1) Essential iris atrophy (1) Hyperopia (2) Left eye blindness (1) Myopia (2) Nystagmus (1) Strabismus (1)	**4** Astigmatism (1) Hyperopia (2) Keratoconus (1) Myopia (1) Strabismus (1)	**7** Amblyopia (1) Astigmatism (3) Hyperopia (1) Myopia (4)	**7** Astigmatism (4) Hyperopia (3) Myopia (1) Nystagmus (1) Strabismus (1)
**Auditory impairment (AI)**	18	4	4	4	6
**Double impairment (VI and AI)**	12	2	2	3	5
**Raven’s Advanced Progressive Matrices—Set 1**	4.9 ± 2.8	4.4 ± 2.8	4.8 ± 3.1	5.4 ± 3.0	4.7 ± 2.2

* Number of participants with a VI highlighted in bold. Participants had either a single or multiple VIs; therefore, the number of VI conditions shown in parentheses does not equal the number of participants with VI.

**Table 2 ijerph-20-07121-t002:** Participants’ self-reported PA levels prior to the start of the MinDSets study.

	Total (*n* = 83)	EXE (*n* = 22)	COG (*n* = 19)	COM (*n* = 21)	CON (*n* = 21)
**Time spent sitting or reclining on a typical day**					
Time (min)	425.8 ± 214.8	418.6 ± 184.8	435.8 ± 189.3	462.9 ± 287.5	387.1 ± 187.8
**Vigorous PA in day-to-day life (e.g., carrying/lifting heavy loads, household chores, etc.)**					
No. of yes responses	14	1	2	4	7
How many days per week? (days)	4.4 ± 1.4	4.0	4.5 ± 2.1	4.3 ± 1.5	4.4 ± 1.6
How much time per day? (min)	39.6 ± 41.7	15.0	45.0 ± 21.2	30.0 ± 0	47.1 ± 58.8
How much time per week? (min)	199.3 ± 308.6	60.0	180.0 ± 0.0	127.5 ± 45.0	265.7 ± 439.7
**Moderate PA in day-to-day life (e.g., brisk walking, carrying light loads, etc.)**					
No. of yes responses	52	16	9	14	13
How many days per week? (days)	4.5 ± 1.5	4.3 ± 1.4	3.9 ± 1.5	4.9 ± 1.3	4.8 ± 1.6
How much time per day? (min)	83.2 ± 171.8	94.1 ± 216.7	70.0 ± 87.6	55.7 ± 46.3	108.5 ± 239.8
How much time per week? (min)	404.6 ± 866.5	454.1 ± 1090.4	295.6 ± 455.1	295.7 ± 278.8	536.5 ± 1199.1
**Walk or use a bicycle to get to and from places for at least 10 min continuously**					
No. of yes responses	41	13	6	10	12
How many days per week? (days)	4.5 ± 1.4	4.7 ± 1.8	3.7 ± 1.2	5.1 ± 1.6	4.2 ± 1.1
How much time per day? (min)	43.8 ± 39.6	52.5 ± 60.7	30.8 ± 18.0	42.5 ± 21.0	42.2 ± 32.2
How much time per week? (min)	186.4 ± 127.4	192.1 ± 102.9	122.5 ± 105.2	233.5 ± 173.1	168.9 ± 107.8
**Participation in sports, fitness or recreational (leisure) activities**					
No. of yes responses	65	16	15	17	17
Type of activities (no. of responses)	Aquatic and paddle sports (e.g., canoeing, diving, kayaking, water polo, swimming etc.) (27), Athletics (track and field) (10), Ball-over-net game (e.g., badminton, tennis, volleyball, table tennis, basketball etc.) (16), Bat-and-ball (e.g., baseball, cricket, softball etc.) (6), Bodyweight exercises (e.g., squats, push-ups, lunges, pull-ups etc.) (21), Bocce (1), Catching games (e.g., dodgeball) (1), Cycling (off-road) (7), Cycling (road) (10), Dancing (28), Equestrian sport (6), Fitness cardio machines (e.g., treadmill, cross-trainer, rowing machine, cycle ergometer) (12), Flying disc sports (e.g., frisbee) (1), Golf (6), Gymnastics (4), Ice sports (e.g., ice hockey, speed skating etc.) (7), Martial arts/combat sports (3), Pilates (2), Powerlifting (1), Running (6), Shooting sports (1), Snow sports (9), Soccer (8), Stick and ball games (e.g., hockey, lacrosse) (3), Target sports (e.g., archery, bowling, curling) (1), Trampoline (1), Walking (40), Weight training (e.g., exercise with heavy weights) (7), Yoga (15), Other: Playing violin (for hours at a time) (1)				
Vigorous intensity sport, fitness or recreational (leisure) activities for at least 10 min continuously					
No. of yes responses	39	8	8	12	11
How many days per week? (days)	3.1 ± 1.5	3.0 ± 1.3	4.1 ± 1.7	2.4 ± 1.3	3.0 ± 1.4
How much time per day? (min)	51.8 ± 31.5	58.1 ± 14.9	63.8 ± 33.8	40.0 ± 15.3	50.0 ± 48.1
How much time per week? (min)	177.3 ± 185.8	176.3 ± 101.7	240.0 ± 133.2	116.4 ± 93.5	195.0 ± 313.4
Moderate intensity sport, fitness or recreational (leisure) activities for at least 10 min continuously					
No. of yes responses	51	13	11	12	15
How many days per week? (days)	3.4 ± 1.8	3.5 ± 2.1	2.8 ± 1.2	3.9 ± 1.7	3.1 ± 1.8
How much time per day? (min)	53.8 ± 50.1	54.2 ± 62.0	68.6 ± 46.6	38.8 ± 23.8	54.7 ± 57.6
How much time per week? (min)	159.6 ± 141.4	175.4 ± 213.9	182.7 ± 135.8	157.5 ± 125.3	130.7 ± 71.6

**Table 3 ijerph-20-07121-t003:** Adherence rates to the eight-week intervention programme for EXE, COG, and COM.

Adherence		EXE (*n* = 22)	COG (*n* = 19)	COM (*n* = 21)
Walks	Total walks completed	26.0 ± 5.2		26.5 ± 4.1
	% of target number (out of 24)	108.1 ± 21.5		110.3 ± 16.9
	Average walks per week	3.3 ± 0.6		3.3 ± 0.5
Cognitive training	Total levels completed		435.5 ± 61.9	443.1 ± 73.3
	% of target levels (out of 384)		113.4 ± 16.1	115.4 ± 19.1
	Average levels per week		53.2 ± 7.0	56.5 ± 7.9
	Sessions completed		45.8 ± 4.8	48.4 ± 5.9
	% of target sessions completed (out of 48)		95.5 ± 10.1	100.8 ± 12.4

**Table 4 ijerph-20-07121-t004:** SART outcomes for GO- and NO-GO-trials shown for all participants combined (total) and split by groups.

GO-Trials	Total		EXE		COG		COM		CON	
	Pre	Post	Pre	Post	Pre	Post	Pre	Post	Pre	Post
Correct response time (ms)	**295.0 ± 79.2 ***	270.5 ± 91.6	298.8 ± 84.4	289.1 ± 101.0	261.5 ± 74.1	241.5 ± 99.2	293.7 ± 75.3	256.3 ± 77.3	322.8 ± 75.9	291.3 ± 83.7
Number correct (/200)	**182.3 ± 27.8 ^#^**	**188.8 ± 23.9**	**186.1± 12.4 ^#^**	**190.5 ± 13.3**	**191.7 ± 8.2 ^#^**	**197.5 ± 4.1**	**182.0 ± 29.1 ^#^**	**193.8 ± 8.9**	**170.3 ± 43.0 ^#^**	**174.2 ± 41.7**
Number incorrect (/200)	**17.7 ± 27.8 ^#^**	**11.2 ± 23.9**	**13.9 ± 12.4 ^#^**	**9.5 ± 13.3**	**8.3 ± 8.2 ^#^**	**2.5 ± 4.1**	**18.0 ± 29.1 ^#^**	**6.2 ± 8.9**	**29.7 ± 43.0 ^#^**	**25.8 ± 41.7**
**NO-GO-Trials**	**Total**		**EXE**		**COG**		**COM**		**CON**	
	**Pre**	**Post**	**Pre**	**Post**	**Pre**	**Post**	**Pre**	**Post**	**Pre**	**Post**
Incorrect response time (ms)	**286.5 ± 142.0 ***	222.6 ± 117.2	**278.0 ± 126.3 ***	210.7 ± 121.4	233.4 ± 146.2	193.8 ± 98.8	**274.5 ± 113.5 ***	216.4 ± 104.3	343.8 ± 163.9	264.1 ± 135.5
Number correct (/25)	**14.6 ± 5.8 ^#^**	**16.7 ± 5.7**	**15.3 ± 6.6 ^#^**	**18.4 ± 5.4**	**14.5 ± 5.7 ^#^**	**16.7 ± 5.7**	**14.1 ± 6.0 ^#^**	**15.8 ± 5.5**	14.5 ± 5.4	15.9 ± 6.1
Number incorrect (/25)	**10.4 ± 5.8 ^#^**	**8.3 ± 5.7**	**9.7 ± 6.6 ^#^**	**6.6 ± 5.4**	**10.5 ± 5.7 ^#^**	**8.3 ± 5.7**	**10.9 ± 6.0 ^#^**	**9.2 ± 5.5**	10.5 ± 5.4	9.1 ± 6.1

* indicating a significant difference between pre- and post-intervention measurements. ^#^ showing positive significant interaction highlighted by Chi-square analysis. All significant findings have also been highlighted in bold font.

**Table 5 ijerph-20-07121-t005:** STROOP outcomes for compatible and incompatible trials for all participants combined (total) and split by groups.

Compatible	Total		EXE		COG		COM		CON	
	Pre	Post	Pre	Post	Pre	Post	Pre	Post	Pre	Post
No. of trials	59.5 ± 5.3	60.1 ± 1.2	58.4 ± 6.2	60.0 ± 0.0	60.3 ± 4.8	60.2 ± 2.2	60.0 ± 3.3	60.0 ± 0.0	58.9 ± 6.1	60.3 ± 1.3
Response time (ms)	**1096.2 ± 232.2 ***	**1053.3 ± 224.0**	**1011.5 ± 180.9 ^§^**	1007.2 ± 204.5	**1013.8 ± 181.5 ^§^**	927.5 ± 180.9	1140.6 ± 270.6	1099.6 ± 166.0	**1221.1 ± 227.1**	1174.9 ± 267.3
**Per cent correct ^§^**	**84.9 ± 22.0 ^#^**	**90.8 ± 16.3**	**94.7 ± 8.7 ^#^**	**96.4 ± 6.3**	**86.3 ± 22.8 ^#^**	**94.7 ± 9.7**	**82.3 ± 23.0 ^#^**	**89.4 ± 21.1**	**79.7 ± 20.4 ^#^**	**84.0 ± 19.9**
Per cent incorrect	**3.9 ± 9.8 ^#^**	**2.9 ± 6.2**	**2.3 ± 6.4 ^#^**	**1.9 ± 4.8**	**6.5 ± 14.0 ^#^**	**3.8 ± 8.1**	**4.8 ± 12.5 ^#^**	**3.8 ± 8.0**	**2.5 ± 2.7 ^#^**	**2.2 ± 2.9**
**Per cent timeouts ^§^**	**11.2 ± 18.8 ^#^**	**6.3 ± 12.8**	**3.1 ± 3.8 ^#^**	**1.7 ± 3.0**	**7.2 ± 13.4 ^#^**	**1.5 ± 3.0**	**13.0 ± 19.8 ^#^**	**6.8 ± 14.2**	**17.8 ± 19.2 ^#^**	**13.8 ± 18.2**
**Incompatible**	**Total**		**EXE**		**COG**		**COM**		**CON**	
	**Pre**	**Post**	**Pre**	**Post**	**Pre**	**Post**	**Pre**	**Post**	**Pre**	**Post**
No. of trials	180.5 ± 5.3	179.9 ± 1.2	181.6 ± 6.2	180.0 ± 0.0	179.7 ± 4.8	179.8 ± 2.2	180.0 ± 3.3	180.0 ± 0.0	181.2 ± 6.1	179.7 ± 1.3
Response time (ms)	1188.1 ± 241.8	1159.0 ± 226.1	**1119.7 ± 190.3 ^§^**	1141.3 ± 209.0	1127.4 ± 188.7	1047.8 ± 197.5	1194.2 ± 313.7	1189.4 ± 186.8	**1318.5 ± 215.1**	1252.7 ± 271.7
**Per cent correct ^§^**	**67.1 ± 34.3 ^#^**	**79.1 ± 26.9**	**83.5 ± 25.0 ^#^**	**88.7 ± 19.0**	**69.5 ± 33.6 ^#^**	**84.2 ± 23.9**	**62.6 ± 34.2 ^#^**	**81.1 ± 22.6**	**54.8 ± 36.7 ^#^**	**65.2 ± 32.6**
Per cent incorrect	**16.9 ± 23.6 ^#^**	**11.4 ± 18.4**	**11.5 ± 23.3 ^#^**	**7.4 ± 16.7**	**15.2 ± 19.0 ^#^**	**12.7 ± 22.6**	**19.7 ± 26.2 ^#^**	**8.8 ± 11.4**	**22.6 ± 25.4 ^#^**	**16.3 ± 21.3**
**Per cent timeouts ^§^**	**16.0 ± 21.5 ^#^**	**9.4 ± 14.7**	**5.0 ± 5.1 ^#^**	**4.0 ± 4.3**	**15.3 ± 23.7 ^#^**	**3.1 ± 4.9**	**17.7 ± 20.7 ^#^**	**10.1 ± 15.6**	**22.7 ± 20.3 ^#^**	**18.5 ± 19.0**

^§^ indicates a significant difference at baseline to CON for response times or a significant difference at baseline between groups for percentage correct responses and timeouts. * highlighting a significant difference between pre- and post-intervention measurements. ^#^ showing positive significant interactions highlighted by Chi-square analysis. All significant findings have also been highlighted in bold font.

## Data Availability

Data available on request due to ethical restrictions. The data presented in this study are available on request from the corresponding author. The data are not publicly available due to ethical restrictions.

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
