# Peer review of "Impact of Prescribed Exercise on the Physical and Cognitive Health of Adults with Down Syndrome: The MinDSets Study"

_ijerph, 2023, doi:10.3390/ijerph20237121_

Round 1

Reviewer 1 Report

TITLE AND ABSTRACT

Title:

The title of the article, "Impact of Prescribed Exercise on the Physical and Cognitive Health of Adults with Down Syndrome: The MinDSets Study," is informative and effectively conveys the subject of the research. It highlights the key elements of the study, including the intervention (prescribed exercise), the target population (adults with Down Syndrome), and the outcomes of interest (physical and cognitive health).

Abstract:

The abstract provides a concise and clear overview of the study, summarizing its purpose, methods, key findings, and implications.

INTRODUCTION

Clarity and Background:

The introduction of the article provides a clear background on Down Syndrome (DS) and its associated health challenges. It effectively explains the genetic basis of DS (Trisomy-21) and mentions the various physiological and cognitive alterations associated with the condition.

Relevance to the Study:

The introduction appropriately sets the stage for the study by highlighting the importance of addressing health issues in the DS community. It emphasizes the need to investigate the impact of prescribed exercise on the physical and cognitive health of adults with DS, which aligns with the study's objective.

Mention of Previous Research:

The introduction references previous research findings to provide context. It mentions studies that have explored exercise interventions in the DS population and highlights the limited number of well-controlled studies, which emphasizes the gap that the current study aims to fill.

Clear Objectives and Hypotheses:

The introduction clearly states the objectives of the study, focusing on the effects of an 8-week walking intervention on physical and cognitive functioning in adults with DS. It also presents a well-defined hypothesis that walking will lead to meaningful improvements in both physical and cognitive functioning, surpassing those seen with cognitive-based training alone.

Flow and Transition:

The introduction maintains a logical flow of information, transitioning smoothly from background information to the specific objectives of the study. However, it could benefit from a brief overview sentence or two that summarizes the key points of the introduction before moving into the methodology section.

METHODS

Ethical Considerations:

The methodology section appropriately mentions that the study was approved by the Anglia Ruskin University Research Ethics Committee and conducted following the Declaration of Helsinki guidelines, demonstrating adherence to ethical standards.

Study Design:

The study design, a repeated-measures, matched-groups design, is well-suited for investigating the effects of interventions on a specific population. Assigning participants to different groups based on pre-intervention measures is a robust method for controlling confounding variables.

Participant Recruitment:

The section provides clear criteria for participant inclusion and exclusion, such as age, ambulatory status, medical clearance, and access to necessary devices. However, it would be beneficial to include information about the recruitment period and location.

Participant Demographics:

The methodology includes a breakdown of participant characteristics by group, providing valuable demographic information. This transparency is important for assessing the generalizability of the study's findings.

Sample Size Determination:

The methodology mentions that group sizes were determined using established criteria, which is good practice. However, it would be helpful to explicitly state the total number of participants in each group.

Data Collection:

The section explains the use of Fitbit devices to collect physical activity data and specifies that participants were provided with instructions and online support. This ensures data collection consistency.

Cardiorespiratory fitness assessment through the six-minute walk test (6MWT) is a recognized method, and the protocol is adapted from previous research [27]. This provides a basis for comparability with other studies.

Cognitive and executive function assessments are conducted using recognized online platforms (PsyToolkit and BrainHQ) and include detailed descriptions of the tests and instructions for participants.

The methodology clearly describes how cognitive and physical activity data were collected, monitored, and transferred for analysis.

Statistical Analysis:

The statistical analysis section outlines the software (SPSS) used for data analysis and mentions the tests and post-hoc procedures applied. It's important to maintain transparency in reporting statistical methods.

Normality testing and tests for homogeneity of variance are mentioned, which is crucial for assessing the validity of statistical tests.

The section provides information about how non-parametric data were analyzed, demonstrating flexibility in the statistical approach.

Clarity and Organization:

The methodology is generally well-organized and clear, with subsections for each aspect of data collection and analysis. However, it can be somewhat lengthy and detailed. Consider summarizing complex procedures when possible to enhance readability.

Reproducibility:

Including references to the specific versions of software and clear descriptions of protocols for physical and cognitive assessments enhances the study's reproducibility.

RESULTS

Clarity and Organization: The text is generally clear and well-organized, making it easy for readers to follow the results. The use of tables and figures aids in presenting data effectively.

Baseline Measurements: The text starts by describing the baseline measurements of various parameters, including physical fitness and cognitive function. This is essential for understanding the starting point of the participants.

Statistical Analysis: The text reports statistical analyses, which is crucial for scientific validity. However, it lacks specific details about the statistical tests used, such as the exact names of tests (e.g., t-tests, ANOVA), degrees of freedom, and effect sizes, which would improve transparency.

Adherence Rates: Adherence rates for the intervention are presented, but there is no critical analysis or discussion of these rates. A discussion of why participants may not have adhered to the intervention and how it might affect the results would add depth to the interpretation.

Physical Fitness Outcomes: The text reports an increase in total distance covered in the 6-minute walk test (6MWT) after the intervention, with significant improvements in the exercise (EXE) and combined (COM) groups. This information is valuable, but it lacks context. For instance, is this improvement clinically significant? How does it compare to previous studies or established norms?

Chi-Square Analysis: The text mentions significant interactions in the Chi-square analyses for some outcomes. However, it does not provide a clear explanation of what these interactions mean or their relevance to the study's objectives. More interpretation is needed.

Comparison Between Groups: The text highlights differences between groups in some cognitive outcomes but notes that these differences were not significant when baseline differences were considered. It would be helpful to discuss the potential impact of these baseline differences on the results.

DISCUSSION

Clarity and Organization: The discussion is generally clear and well-structured. It effectively outlines the study's hypothesis and provides a comprehensive explanation of the results. However, some paragraphs are quite long, which could be improved for readability.

Hypothesis and Acceptance: The discussion opens by restating the study's hypothesis and concludes that the findings support this hypothesis. This is an appropriate way to start the discussion, as it clearly communicates the study's aims and outcomes.

Explanation of Walking as a Cerebral Modulator: The text explains why walking is considered a cerebral modulator in the DS population, emphasizing the complex cognitive and motor processes involved. However, it would be beneficial to include references or citations to support these claims, especially when referring to specific brain pathways and cognitive processes.

Importance of Cognitive Load in Walking: The discussion argues that walking applies a cognitive load, even though it may seem minimal for the general population. This is an interesting point, but it would be valuable to provide more context on why this cognitive load is more significant for individuals with DS, especially those with impaired cognitive processing capabilities.

Interpretation of Cognitive Test Results: The discussion provides a detailed interpretation of the results for cognitive tests such as the SART and STROOP. It correctly links the improvements in cognitive outcomes to walking, highlighting the role of attention and vigilance. However, the discussion could benefit from more context on the practical implications of these cognitive improvements for individuals with DS.

Comparison Between Groups: The text discusses the differences between the exercise (EXE), cognitive (COG), combined (COM), and control (CON) groups, emphasizing the unique benefits of each intervention. It would be helpful to acknowledge the limitations of comparing these groups, such as potential confounding variables or differences in intervention intensity.

Limitations: The discussion appropriately addresses the study's limitations, including data collection methods, duration of the intervention, and potential weather-related impacts. However, it would be beneficial to discuss how these limitations might have influenced the results and how they could be addressed in future research.

Future Directions: The text mentions the need for future research, particularly in exploring more complex locomotive activities and the impact of exercise intensity. This is a valuable suggestion and aligns with the study's findings. However, providing more specific research questions or directions for future studies would enhance this section.

CONCLUSIONS

Clarity and Conciseness: The conclusion starts well by summarizing the study's significance. However, it could benefit from more concise wording. For example, "These responses were magnified through a combined dose of exercise and cognitive training" could be simplified for better clarity.

Practical Implications: While the conclusion mentions the importance of the findings and their potential impact on the DS community, it could delve deeper into the practical implications. How might these cognitive improvements benefit individuals with DS in their daily lives? Are there specific interventions or strategies that can be recommended based on these findings?

Future Directions: While the conclusion briefly mentions the potential for adopting exercise and cognitive training, it could expand on this by suggesting specific avenues for future research or initiatives. What should researchers or practitioners focus on next? Are there unanswered questions that need further exploration?

Author Response

Please refer to the attached file for a point-by-point response to all of the comments that were raised. 

Reviewer 2 Report

Dear authors,

Thank you for the possibility to read the interesting article. 

The article is well performed and well written. However one objections is that although the  limitations of the study are obvious and highlighted, the following conclusion is a bit exessive. The short project period, few individuals in some groups, the mode of collecting data are examples of weaknesses that could hold back the conclusions a bit.   

Author Response

(The authors gave the same response as above.)

Reviewer 3 Report

I find the article very interesting as there is a lot of research on Down syndrome interventions in childhood, but not in adulthood.

The background seems to me to be very well written but perhaps it could be interesting in terms of how the methodology is elaborated and what is also provided in the discussion, to create a context for telerehabilitation.

In the section on material and methods, it would be important to explain how the ethical criteria are maintained in the CON group (without a therapeutic intervention even assuming the hypothesis of benefits). Also, in addition to the inclusion criteria, it should be stated whether exclusion criteria were considered. To clarify this section, a flow chart according to Consort standards could be incorporated.

The inclusion of the tables with the socio-demographic variables and sensory impairments in the patients is very interesting, although it would be necessary to explain the adaptations made to the software used in the case of visual and/or hearing impairment (for example, in BrainHG), as well as to explain the technical details of the devices and applications.

The importance of a caregiver to support the motivation and use of devices is emphasised, but this fact and its importance is not highlighted in the rest of the sections of the study.

The tools and scales are explained in detail, as well as all the statistical procedures carried out.

As for the results section, the graphs and tables are enlightening and rigorous.

Finally, the discussion section emphasizes the neurophysiological basis of gait and its relationship with cognitive processes, but it would also be interesting to include the impact that the caregiver and the use of informational technology may have had on motivation and adherence.

Limitations include the fact that there were no laboratory conditions, and it is a strength to recognise weaknesses, but it would be interesting to highlight how this project has been guided by professionals and could be framed in telerehabilitation programmes. Moreover, it could be a cost-benefit therapy that could be assumed by the community.

Although a prospective is outlined, some clearer aspects could be included. It is suggested that perhaps 8 weeks is too short a time. In this sense, in future research, how long would the programme be maintained over time? What other improvements would be considered?

Author Response

(The authors gave the same response as above.)

Reviewer 4 Report

I thank for the opportunity to review this manuscript. It presents an innovative study about the benefits of exercise in the cognitive, executive, and physical health of adults with DS. The introduction presents a very clarifying rationale of the study.  The objectives are clearly stated. The methods are reported in excellent detail. 

However, I have a few concerns highlighted bellow: 

Abstract:

Line 22. “from pre- to post-intervention in the NO-GO-trials (p < 0.05) across all groups.” Please, explain the mean of No-Go trial, if not, this outcome is not easily understood in the context of the abstract.  Please, correct across all groups, it is not like this. The interaction is not significant in control group. 

In the results, in the abstract it might be better to state positive significant interactions. 

Correct the conclusion according to the indications for the main text. 

Methods

Line 103. “participants were assigned to”. Maybe it is better to explain that the participants were stratified according to pre-intervention measures of cardiorespiratory fitness in order to maintain a similar level of cardiorespiratory fitness between the participants in the different groups. 

Table 1. Please, reformat the table according to journal specifications. 

Results

Tables. Please, reformat and renumber the tables according to journal specifications. For example, there are two tables 1, the second one, is referenced in the text as table 3. 

Statistical analysis

Which test was used to non-parametric post-intervention differences between groups? I suppose that according to this sentence “Time-by-group interactions across all measures and groups were determined using repeated-measures ANOVA.” ANOVA, considering Levene’s test was used for parametric and non-parametric data, please confirm. 

5. Conclusions

Line 545: “Through the simple application of walking, a form of exercise which requires little to no equipment or expense, there were significant increases in cognitive and executive function, reflecting improved capabilities in key attributes of information processing, vigilance, selective attention, processing speed, and short-term memory.”

 I think the capabilities processing speed and short-term memory might not be included in the conclusion section. Statistical results of ANCOVA for processing speed and statistical results of CORSI for short-term memory are not conclusive. 

Author Response

(The authors gave the same response as above.)

Reviewer 5 Report

General questions

The manuscript has merit because it has interesting and relevant results on functional and cognitive adaptations of people with Down Syndrome undergoing training. It has adequate theoretical justification, design, statistical analysis and discussion. However, some items need to be corrected and clarified, especially in the results.

 Major questions

- Table 4: the size of the * and # signs are too small. It is difficult to make the distinction.

- interactions from Chi-square analysis: In Table 4 it is represented by # and in Figure 3 it is represented by *. It is more appropriate to standardize the signs independently of the table and/or figure. (See Table 5 and Figure 4, too).

- Table 5: the data was disconfigured

Author Response

(The authors gave the same response as above.)

Round 2

Reviewer 1 Report

The authors have made the necessary changes, and the article has improved its quality.